# Ecological Significance of a Novel Nitrogen Fixation Mechanism in the Wax Scale Insect *Ericerus pela*

**DOI:** 10.3390/insects16080836

**Published:** 2025-08-13

**Authors:** Qian Qi, Bin Li, Xin Zhang, Xiaoming Chen, Hang Chen, Kirst King-Jones

**Affiliations:** 1Forest Institute of High Land, Chinese Academy of Forestry, Kunming 650224, China; qiqian@szbl.ac.cn (Q.Q.);; 2Research Center of Resource Insects, Chinese Academy of Forestry, Kunming 650224, China; libin2015caf@126.com; 3Yunnan Key Laboratory of Breeding and Utilization of Resource Insects, Kunming 650224, China; 4Department of Biological Sciences, University of Alberta, G-504 Biological Sciences Bldg., Edmonton, AB T6G 2E9, Canada

**Keywords:** the wax scale insect *Ericerus pela*, host plant overly dependent, obligate symbiotic microorganisms, nitrogen fixation, amino acid synthesis

## Abstract

*Ericerus pela* is a small insect that lives on plants and feeds on their sap, but plant sap has very little nitrogen—essential for growth. The question arises of how these insects get enough nitrogen. Scientists studied this by testing if they can turn nitrogen from the air into a usable form (a process called nitrogen fixation). We found female insects are good at this. Further research showed their bodies host 42 types of bacteria that help with nitrogen fixation, like common bacterial groups *Rhizobiales* and *Methylobacterium*. Genes involved in using nitrogen, moving it around, and making amino acids were active at different life stages. Additionally, a fungus living in their bodies can make all the amino acids they need, including those they cannot make themselves. This means *E. pela* relies on both bacteria (to fix nitrogen) and fungi (to make amino acids) to get enough nitrogen. This discovery shows how the insect evolved to work with microbes to thrive on low-nitrogen food. Understanding this could help us learn more about how living things team up with microbes to solve nutrient problems.

## 1. Introduction

*Ericerus pela*, commonly known as the white wax scale insect (WSI), is widely found in Asia and Europe. The white wax secreted by these insects has been broadly used for thousands of years in food, medical, cosmetic, and traditional wax printing industries, providing significant economic, medical, and cultural value to humans [1,2]. This unique biological trait not only underlies their profound importance to humans, but also reflects a remarkable evolutionary adaptation to their highly specialized and constrained lifestyle. While the epidermis of female *E. pela* thickens into a protective cuticular scale that exceeds their original body size by roughly fourfold, males produce large amounts of wax secretions (≈30% body mass) that completely envelop their bodies [3], resulting in permanent attachment to the host plant. This sessile, wax-encased lifestyle renders these insects exclusively reliant on phloem sap from the host plant as their sole nutritional source. However, the low nitrogen (N) content in phloem sap—which typically ranges from only 0.1 to 2 μg μL^−1^ [4]—poses a critical challenge for their survival. The mechanisms by which *E. pela* obtains sufficient nitrogen to sustain development and reproduction remain poorly understood and present an intriguing biological mystery.

Phloem sap-feeding insects (PFSIs) commonly rely on symbiotic microorganisms for amino acids synthesis, thereby mitigating nitrogen scarcity. To meet their specific nutritional requirements, different PSFIs have evolved specialized symbiont systems. For instance, in aphids, the primary symbiont *Buchnera* is responsible for synthesizing essential amino acids [5,6,7]. Similarly, other PSFI species also rely on distinct microbial partners. The planthoppers *Laodelphax striatellus* and *Nilaparvata lugens* associate with fungal symbionts like *Cordyceps* or *Candida* [8,9,10], while the leafhopper *Hishimonus phycitis* harbors Hp-YLS and *Candida pimensis* [11]. In addition, scale insects such as *Kermes quercus* and cicadas have been reported to associate with *Ophiocordyceps* [11,12]. These diverse partnerships highlight the evolutionary importance of symbiotic microorganisms in enabling PFSIs to survive and thrive despite maintaining nitrogen-poor diets.

Nitrogen is one of the most critical elements for amino acid synthesis in symbiotic microorganisms, yet phloem sap is notoriously nitrogen-poor and insufficient to meet this demand as the sole food source in PFSIs. To overcome this nitrogen deficit, atmospheric nitrogen fixation provides an alternative strategy for obtaining supplemental nitrogen substrates for the biosynthesis of amino acids in the symbiotic microorganisms of PFSIs. Nitrogen fixation in insects was first discovered in termites [13,14] and later observed in beetles [15,16,17], both of which primarily feed on plant xylem, a nutrient-poor substrate with limited nitrogen. To compensate for this nutritional shortage, they acquire nitrogen-fixing bacteria horizontally through their diet as opposed to vertically transmitted symbionts from their parents [17,18]. Consequently, this facultative symbiotic relationship is inherently unstable and depends highly on the dietary composition of the host insects.

The exact mechanisms of symbiotic nitrogen fixation in insects and their associated symbionts remain unclear, whereas those in plants have been extensively studied and well-characterized. Plant roots secrete flavonoids and other chemoattractants (e.g., lipo-chitooligosaccharides) to recruit nitrogen-fixing bacteria *Rhizobia*, which reduce N_2_ to NH_3_ in anaerobic microenvironments via nitrogenase that contains an iron–molybdenum cofactor. Ammonium is transported across cell membranes through multiple approaches, such as ammonium transporters (AMT) [19]. Plant cells then assimilate ammonium into glutamine, which is transported to aerial tissues via the xylem. Nitrogenase of *Rhizobia* is encoded by the *nif* gene family, with *nif*H being highly conserved and widely used for identifying nitrogen fixing bacteria. Recent studies even demonstrate that a yeast-like organism within cyanobacteria evolves into a specialized nitrogen-fixing organelle, which has been termed nitroplast [20].

To unravel the potential mechanisms underlying nitrogen acquisition in the phloem-feeding white wax scale insect *E. pela*, we hypothesize that symbiotic microorganisms play a critical role in this process. First, to elucidate the endogenous nitrogen fixation activity within *E. pela*, we employed stable isotope tracing techniques combined with the acetylene reduction assay (ARA) to qualitatively and quantitatively characterize its nitrogen fixation capacity. To further elucidate the microbial basis of nitrogen acquisition, we systematically investigated the diversity, localization, and functional roles of symbiotic microbes involved in nitrogen fixation and amino acid biosynthesis, integrating multi-omics approaches (genome, metagenome, and transcriptome analyses) to delineate putative nitrogen fixation and assimilation pathways within the insect host. This study aims to advance our understanding of nitrogen fixation mechanisms in resource insects and forestry pests adapted to low-nitrogen diets, thereby providing novel insights for the integrated management of agricultural and forestry pests, and contributing to the conservation of forest and agricultural ecosystem health.

## 2. Materials and Methods

### 2.1. Sample

*E. pela* used in this research was collected on the host plant *Ligustrum lucidum* at the experimental garden of Kunming, Yunnan Province, China (N 25°3′47″, E 102°45′18″).

### 2.2. Identification of Azotobacter in E. pela

Isolation and Culture of Azotobacter. The *E. pela* was immersed in alcohol and distilled water, and was cleaned three times in an ultrasonic cleaner (OEM, JPS-100A, Shenzhen, China). The surface of the *E. pela* female adult was washed three times with 75% alcohol, and then washed three times with distilled water. The intact carcasses, epidermis, and digestive tract were homogenized with PBS and cultured on nitrogen-deficient HW medium [21] (mannitol 10.0 g/L, potassium dihydrogen phosphate 0.2 g/L, magnesium sulfate heptahydrate 0.2 g/L, sodium chloride 0.2 g/L, calcium sulfate dihydrate 0.2 g/L, calcium carbonate 5.0 g/L, agar 20.0 g/L, pH 7.0–7.2 at 25 °C).

16S rDNA Sequencing. Microbial community genomic DNA was extracted from WSI samples using the E.Z.N.A.^®^ soil DNA Kit (Omega Bio-tek, Norcross, GA, USA) according to the manufacturer’s instructions. For DNA extraction, the following quantities of *E. pela* were used: 100 1st-instar male and female nymphs; 30 2nd-instar male and female nymphs, 30 male pupae, and 30 male adults; 10 female adults. Each group included 3 biological replicates to ensure statistical robustness. The hypervariable regions V3–V4 of the bacterial 16S rRNA gene were amplified with primer pairs 341F (5′- CCTAYGGGRBGCASCAG-3′) and 806R (5′-GGACTACNNGGGTATCTAAT-3′). The hypervariable regions of the fungal ITS gene were amplified with primer pairs 1737F (5′-GGAAGTAAAAGTCGTAACAAGG-3′) and 2043R (5′-GCTGCGTTCTTCATCGATGC-3′). The raw 16S rRNA gene sequencing reads were demultiplexed, quality-filtered in fastp version 0.20.0 [22] and merged in FLASH version 1.2.7 [23]. Operational taxonomic units (OTUs) with 97% similarity cut off were clustered using UPARSE version 7.1 [24], and chimeric sequences were identified and removed. The taxonomy of each OTU representative sequence was analyzed by RDP Classifier version 2.2 [25] against the 16S rRNA database (e.g., Silva v138) using a confidence threshold of 0.7.

### 2.3. Fluorescence In Situ Hybridization

Fluorescence in situ hybridization was performed following previously described methods [26]. According to the 16S rDNA characteristic sequence of *Ophiocordyceps sinensis*, a fluorescent probe was designed and synthesized according to the 16S rDNA characteristic sequence of *Ophiocordyceps sinensis*: 5′-CAGGCACGTCAGCGCTCG-3′. The paraffin section specimen was dehydrated with graded alcohol and hydrated with DEPC-PBST for 1 h. Then, 6% H_2_O_2_ was prepared with DEPC-PBST, and after bleaching the samples on ice for 1 h, they were washed 3 times with DEPC-PBST for 5 min. After permeabilization with 20 mg/mL Proteinase K for 8 min at room temperature, samples were washed 3 times with DEPC-PBST for 5 min and then fixed with PFA at room temperature for 20 min. The samples were again washed 3 times with DEPC-PBST for 5 min, and then the sample was incubated in pre-hybridization solution at 68 °C for 60 min; the probe was diluted in pre-hybridization solution, and the sample was subsequently incubated at 68 °C for 48 h. Washing steps were performed sequentially: three washes with L1 Buffer at 68 °C for 30 min each, followed by three washes with L2 Buffer (50% Formamide, 2XSSC, pH 4.5, 0.1% Tween 20) with the same temperature and duration. Subsequently, three additional washes were conducted with L3 Buffer (2XSSC, pH 4.5, 0.1% Tween 20) at 68 °C for 10 min per wash cycle. The samples were then treated with a 1:1 mixture of L3 Buffer and RNase Buffer [composed of 500 mM NaCl, 10 mM Tris (pH 7.5), and 0.1% Tween 20] at room temperature for 5 min. RNase digestion was carried out by incubating specimens in 100 μg/mL RNase solution at 37 °C for 1 h. Final washing steps involved three sequential rinses with Tris/NaCl solution [containing 8 g/L NaCl, 0.2 g/L KCl, 3 g/L Tris (pH 7.4), and 1% Tween 20] at room temperature, with each wash lasting 5 min. We then diluted the probe in the pre-hybridization solution, incubated the sample at 68 °C for 48 h, washed it 3 times with L1 Buffer at 68 °C for 30 min each time, washed it 3 times with L2 Buffer (50% Formamide, 2XSSC, pH 4.5, 0.1% Tween 20) at 68 °C for 30 min each time, and washed it 3 times with L3 Buffer (2XSSC, pH 4.5, 0.1%Tween 20) at 68 °C for 10 min each time. We then washed with a 1:1 mixture of L3 Buffer and RNase Buffer (RNase Buffer: 500 mM NaCl, 10 mM Tris pH 7.5, 0.1% Tween 20) at room temperature for 5 min. Then, we incubated the specimen with 100 ug/mL RNase solution at 37 °C for 1 h. After this, we washed it 3 times with Tris/NaCl solution (Tris/NaCl: 8 g/L NaCl, 0.2 g/L KCl, 3 g/L Tris pH 7.4 1% Tween 20) at room temperature, 5 min each time. The specimen was incubated with DAPI for 10 min. Then, the samples were observed and photographed with a fluorescence microscope (NIKON ECLIPSE CI, Nikon Corporation, Tokyo, Japan).

### 2.4. Stable Isotope Labeling Test

Stable isotope labeling was performed following previously described methods [27]. For this procedure, 100 live second-instar female nymphs of the *E. pela* were put into a 50 mL syringe, the plunger was pushed close to the bottom, 6 mLO_2_ and 24 mL ^15^N_2_ were sucked into the syringe, and the air inlet was immediately sealed with heated wax. After 24 h, the WSIs were collected and put into an oven at 70 °C for 72 h in order to remove free ^15^N_2_. After the sample was fully burned, N_2_ was formed and analyzed in an elemental analyzer. The content of ^15^N and ^14^N in the N_2_ was detected by use of a mass spectrometer. The ratio was compared with the international standard (Atm-N_2_) to calculate the d^15^N value of the sample. Nitrogen content (N%) was calculated by comparing the peak areas of the sample and three working standards. The second-instar female nymphs killed in 75% alcohol are referred to as CK. There were 5 replicates for each sample.

### 2.5. Acetylene Reduction Test

An acetylene reduction test was performed following previously described methods [13]. For this procedure, 100 live second-instar female nymphs of the *E. pela* were put into a 50 mL syringe, the plunger was pushed close to the bottom, 3 mL acetylene and 24 mL fresh air were inhaled into the syringe, and the air inlet was immediately sealed with heated wax. After 48 h, 1 mL of mixed gas was extracted with a gas tight sampling probe, and injected into the gas chromatography probe to detect the content of ethylene generated. Then, 2nd-instar female nymphs killed in 75% alcohol were denoted as CK. There were 5 replicates per sample. The nitrogenase activity is calculated as follows:$$N=\frac{hx\times c\times v}{hs\times 24.9\times t}$$

*N*: Concentration of produced ethylene (nmol/mL·h). *hx*: The peak value of the sample. *c*: Concentration of the standard ethylene (nmol/mL). *v*: The volume of the culture container (mL). *hs*: The peak value of the standard ethylene. *t*: The reaction time (h). 24.9: The volume of standard ethylene at 30 °C (mL).

### 2.6. Whole Genome Sequencing and Analysis

We sequenced the genome for first-instar female nymph of WSI using the Illumina Hiseq 2000 platform (San Diego, CA, USA). The splicing software velvet 1.2.10 was used for the sequence splicing. The Hi-C libraries were sequenced on the Illumina HiSeq X Ten platform in 150 PE mode (Illumina, San Diego, CA, USA), which yielded 87 Gb of data. The sequence comparison software Mummer 4.0.0 was used to align the contig sequences (>1 kb) to the genome sequence of the *E. pela*. Glimmer 3.02 software was used for ORF (open reading frame) prediction, and all predicted protein sequences were aligned to NT (NCBI nonredundant protein sequences), NR (NCBI nonredundant database), KOG (euKaryotic Ortholog Groups), KEGG (Kyoto Encyclopedia of Genes and Genomes), and Swiss-Prot database. BLASTp (E-value < 1 × 10^−10^) alignment was performed with the GO database to the functional annotation of the protein sequence.

### 2.7. Transcriptome Sequencing and Analysis

For RNA extraction, the following quantities of *E. pela* were used: 100 1st-instar male and female nymphs; 30 2nd-instar male and female nymphs, 30 male pupae, and 30 male adults; 10 female adults. Each group included 3 biological replicates to ensure statistical robustness. First-instar nymphs were collected immediately after hatching from eggs, with no prior contact with host plants or feeding, ensuring minimal gut content contamination. Pupae and male adults were collected by peeling off their wax shells and similarly subjected to no feeding prior to sampling, maintaining gut sterility. For second-instar nymphs and female adults, due to their strong attachment to host plants, the samples were quickly desiccated and died upon detachment. Thus, they were used directly without starvation, as prolonged handling would compromise sample integrity. Total RNA was extracted from different developmental stages of WSI by TRIZOL. mRNA was purified from total RNA using magnetic beads coated with oligo-dT. A fragmentation buffer was used to break mRNA into short segments, and these mRNAs were used as templates to synthesize the first strand of cDNA with random hexamers. The second-strand cDNA was synthesized by adding reaction buffer, dNTPs, DNA polymerase I and RNase H, and then was purified by AMPure XP beans. After end repair and a tail and ligation of the sequencing connector, screens were performed with AMPure XP beads. The final library was obtained after PCR amplification and purification. The library quality was assessed on the Illumina HiSeq 2100 platform (BGI, Shenzhen, China). Eighteen transcriptomic libraries (six different developmental stages with three repeats for each) were made in this study. Based on the assembly results, Bowtie2 was used to map the clean reads of each sample to Unigene, and then RSEM was used to calculate the gene expression level of each sample. Blastn, Blastx, and Diamond were used to align Unigenes to the NT, NR, KOG, KEGG, and Swiss-Prot database for annotation. All the genes annotated as nitrogen fixation and amino acid synthesis enzymes were searched out, and the one with the highest transcript abundance was thought to be involved in nitrogen fixation and amino acid synthesis enzymes; their relative expression levels were determined by FPKM (fragments per kilobase of transcript per million fragments mapped).

### 2.8. Metagenome Sequencing and Analysis

Each *E. pela* sample was washed 3 times with 75% alcohol, and ultrasonic cleaning was performed 3 times with distilled water. The cleaned samples were dried and added with 200 μL PBS for grinding to release microorganisms from insect tissues. The insect tissues were filtered out with gauze, and total DNA was extracted from the supernatant. A 350 bp paired end (PE) library was constructed and sequenced using Illumina Hiseq 2000 (Illumina, San Diego, CA, USA). The host gene reads were removed from the database according to the whole-genome data of *E. pela*. The SOAP denovo assembly software was used to assemble the grown contigs from effective reads. Contigs longer than 500 bp were used for ORFs prediction by MetaGene program. Sequences with ORFs longer than 100 bp in each sample were used to construct a non-redundant gene set. SOAP2 was used to compare the effective reads in each sample with the non-redundant gene set (identity ≥ 90%). The number of reads matching the corresponding gene was used to calculate the relative abundance of the gene. Then, the BLASTP program (Expect value < 10^−5^) was used to compare all genes in the gene set with the NR database for homologous classification, and all genes were compared with the KEGG database for functional annotation. Based on the annotation information, the relative abundance of genes was calculated from the same species or the same KO (KEGG portal group), and finally the taxonomy profile and KO profile of microorganisms in the *E. pela* were constructed.

### 2.9. Quantification of mRNA Expression of Target Genes

Quantitative reverse transcription polymerase chain reaction (qRT-PCR) was used to quantify target gene mRNA expression. Total RNA was isolated from about 0.1 mg of each *E. pela* across different life stages using TRIzol reagent (Invitrogen, Waltham, MA, USA), per the manufacturer’s instructions, with DNase I (RNase-free) treatment to remove genomic DNA. RNA quality was assessed by NanoDrop 2000 (Thermo Fisher Scientific, Waltham, MA, USA) and 1.5% denaturing agarose gel electrophoresis. First-strand cDNA was synthesized using PrimeScript™ RT Master Mix (TaKaRa, Kyoto, Japan) with Oligo (dT)18 and random hexamers (20 μL reactions: 1 μg RNA, 4 μL 5× Buffer, 1 μL Enzyme Mix, 1 μL Oligo (dT)18 [50 μM], 1 μL Random 6 mers [100 μM], nuclease-free water). Reverse transcription: 37 °C 15 min, 85 °C 5 s, 4 °C hold. qPCR reactions (20 μL) used SYBR^®^ Premix Ex Taq™ II (TaKaRa) with 2 μL cDNA, 0.4 μL of each primer (10 μM), 10 μL 2× SYBR Mix, and 7.2 μL nuclease-free water, run on ABI 7500 Fast (Applied Biosystems, Waltham, MA, USA) with 95 °C 30 s; 40 cycles of 95 °C 5 s and 60 °C 30 s. Melting curve: 95 °C 15 s, 60 °C 1 min, 20 °C/s to 95 °C. Expression was normalized to the geometric mean of stable housekeeping genes (β-actin) and quantified via 2^−ΔΔCt^.

### 2.10. Amplification of nifH Gene

The primers used for nifH gene amplification were *nif*HF (5′-AAAGGYGGWATCGGYAARTCCACCAC-3′) and *nif*HR (5′-TTGTTSGCSGCRTACATSGCCATCAT-3′). The TransGen AP221–02 Kit (TransGen Biotech, Beijing, China) was used for the PCR amplification. PCR products were sent for sequencing after electrophoresis detection.

### 2.11. Effect of Removing Symbiotic Bacteria on the Survival and Development of E. pela

Ten healthy Ligustrum lucidum trees (approximately 0.5 m in height) with similar growth statuses were selected. Each tree was inoculated with more than 200 eggs of *E. pela*. When nymphs hatched into 1st-instar larvae, the treatment group was sprayed with tetracycline at 1000 ppm (parts per million) on the insect body surface, while the control group (CK) was sprayed with the same volume of distilled water. At the start of the experiment, the number of 1st-instar female nymphs (Ti) on each tree in both the CK and treatment groups was recorded. After 9 months, the number of adult females (Ni) in the corresponding groups was counted again. The survival rate of female nymphs in each treatment group was calculated as follows: Survival rate (%) = (Ni/Ti) × 100%. Additionally, 100 adult females were collected from each treatment group, and their weight per 100 individuals was measured.

## 3. Results

### 3.1. Nitrogen Fixation of E. pela

As a scale insect, *E. pela*, which is fixed on branches (Figure 1A) and relies solely on plant sap with extremely low nitrogen content, may also needs nitrogen from the air to supplement its life. To test whether *E. pela* is capable of utilizing atmospheric nitrogen, we exposed insects to ^15^N-labeled nitrogen gas in a hermetically sealed container for 24 h. We found a significant increase in ^15^N in the dry tissue of viable *E. pela* compared to that of ethanol-killed controls, which also served as a bactericide (*p* = 0.0022) (Figure 1B). This result suggests that *E. pela* can incorporate the free nitrogen in the air into stabilized nitrogen in tissues.

Biological nitrogen fixation is catalyzed by the enzyme nitrogenase, which not only reduces N_2_ to NH_3_, but can also reduce other substrates such as acetylene. The acetylene reduction assay (ARA) has long been used as a proxy to measure nitrogenase activity in diazotrophic organisms, either in symbiotic associations or in their free-living state [13,28]. Therefore, in order to measure nitrogen fixation in *E. pela*, we adopted the classic ARA. Here, we injected acetylene into a sealed bottle housing *E. pela* individuals, and then measured ethylene production using gas chromatography (GC) (Figure 1C). We detected robust nitrogenase activity across multiple life stages. Ethylene production rates were 0.082 ± 0.034 nmol/h (♀) and 0.065 ± 0.005 nmol/h (♂) in first instars, 0.16 ± 0.023 nmol/h (♀) and 0.046 ± 0.02 nmol/h (♂) in second instars, and 0.1849 ± 0.0427 nmol/h in adult females (Figure 1D). By comparison, these rates exceed those of nitrogen-fixing termite soldiers (about 0–0.014 nmol/h), but are lower than those of workers (about 4.4–400 nmol/h) [29].

Data are presented as the mean ± standard deviation (SD, *n* = 5). Statistical analysis was performed using an unpaired Student’s *t*-test, with statistical significance indicated by *p* < 0.05 (*), *p* < 0.01 (**) and ns (not significant) for *p* > 0.05.

### 3.2. The Role of Microbial Symbiosis in the Nitrogen Fixation of E. pela

To identify symbiotic bacteria with a potential role in nitrogen fixation, we conducted high-throughput sequencing on the microorganisms released from tissues of *E. pela*. This analysis revealed 42 species of nitrogen-fixing bacteria, with *Rhizobiales* and *Methylobacterium* emerging as the dominant bacterial species (Figure 2A). Importantly, both bacterial taxa were detectable from the embryonic stage through to adulthood (Figure 2B), suggesting a potential obligate symbiotic relationship acquired through vertical transmission.

To examine where these nitrogen-fixing bacteria reside within *E. pela*, we isolated tissue samples from the digestive tract, the epidermis, and intact carcasses. The samples were homogenized, plated on agar media that lacked nitrogen, and observed for microorganismal growth. Interestingly, bacterial colonies grown on the agar plate were only observed from the epidermal and intact carcass tissue samples, but no colony growth was observed from the digestive tract tissue (Appendix A). This pattern is consistent with the hypothesis that these nitrogen fixing bacteria are obtained through maternal transmission, but not from food ingestion.

In order to more directly observe the distribution of nitrogen-fixing bacteria in the body, we used fluorescent in situ hybridization (FISH) with a red luminescent probe to detect the 16S rDNA of two nitrogen-fixing bacteria, *Rhizobiales* and *Methylobacterium*, which were highly abundant in the second-instar female nymphs of *E. pela*. Paraffin sections of tissue isolated from female second-instar nymphs were analyzed by confocal microscopy. Using this approach, we observed a strong fluorescent signal in the fat body (Figure 2C)—a key organ for storing and metabolizing nutrients. These results suggest that *Rhizobiales* and *Methylobacterium* may play important roles in nutrient metabolism, especially for nitrogenous compounds.

To further confirm the involvement of microbial symbiosis in the nitrogen fixation of *E. pela*, we amplified and sequenced the *nifH* gene, which encodes the dinitrogenase reductase subunit—a critical component of the nitrogenase enzyme complex in nitrogen-fixing bacteria and archaea. As expected, we detected the presence of the bacterial *nifH* gene in *E. pela* across all life stages (Figure 2D), which is consistent with the extensive involvement of nitrogen-fixing microbial symbiosis in the growth and development of *E. pela*.

Data are presented as the mean ± standard deviation (SD, *n* = 3).

### 3.3. Removal of Symbiotic Bacteria Affects the Growth and Survival of the Female E. pela

To investigate the importance of nitrogen-fixing bacteria in the growth and development of *E. pela*, we applied a broad-spectrum antibiotic, tetracycline, or distilled water (control) to the cuticle of newly hatched first-instar female nymphs. At this stage, the nymphs possess a tender cuticle, which allows for the easy absorption of the treatment solutions with or without tetracycline. After eight months, we observed the near-complete elimination of the nitrogen-fixing bacteria in the tetracycline-treated insects (Figure 3B). These individuals exhibited a significantly reduced survival rate compared to controls (*p* = 0.0024) (Figure 2C). Moreover, the surviving female insects were significantly smaller in body size and lower in weight (72.17 ± 4.674 mg) than those in the control group, which retained their symbiotic bacteria (206.89 ± 20.21 mg) (*p* = 0.0045) (Figure 2A,D). Together, these results highlight the essential role of nitrogen-fixing bacteria in sustaining the growth and development of *E. pela*, and underscore their contribution to host fitness in nitrogen-limited environments.

Data are presented as the mean ± standard deviation (SD, *n* = 3). Statistical analysis was performed using an unpaired Student’s *t*-test, with statistical significance indicated by *p* < 0.05 (*), *p* < 0.01 (**), *p* < 0.001 (***), *p* < 0.0001 (****) and ns (not significant) for *p* > 0.05.

### 3.4. Amino Acids Synthesis in E. pela

To investigate the molecular mechanism of nitrogen fixation in *E. pela*, we conducted quantitative real-time PCR (qRT-PCR) to measure the transcript abundance of the ammonium transporter (AMT) across developmental stages. In both nymph and adult stages, AMT genes were actively transcribed, with particularly high expression observed in both male and female second-instar nymphs (Figure 4A). This elevated expression suggests that AMT-mediated ammonium transport plays an active role in nitrogen fixation during a critical developmental window, potentially facilitating the assimilation of fixed nitrogen into host metabolic pathways.

Our transcriptomic data reveal that both glutamate synthase and glutamine synthetase were consistently expressed across all developmental stages of *E. pela*, with particularly high expression levels observed during the second instar stage in both male and female individuals (Figure 4B). This expression pattern coincides with the developmental stage with the most abundant nitrogen-fixing bacteria (Figure 2B), suggesting that *E. pela* may primarily utilizes this pathway for ammonia conversion. Notably, glutamate dehydrogenase—which enables the direct conversion of α-ketoglutarate and NH_4_^+^ into glutamate—also exhibited considerable expression throughout all developmental stages. Interestingly, the first-instar nymphs of both sexes displayed relatively higher expressions of glutamate dehydrogenase compared to the other stages tested, which is different from the result for glutamate synthase and glutamine synthetase, indicating a complementary pattern may exist between these two pathways in response to ammonia concentration fluctuations within *E. pela*.

Previous studies demonstrate that the ABC transporter is pivotal in transporting amino acids between cells in both bacteria and eukaryotes [29,30]. Indeed, we also observed elevated levels of the ABC transporter transcript in all developmental stages of *E. pela*, particularly in the nymphs (Figure 4C), which mirrors their considerable need for amino acids intake to support their growth.

To further investigate potential sources of amino acids in *E. pela*, we re-analyzed genomic data previously acquired and sequenced by our research group [1]. Here, we found that *E. pela* possesses a complete set of genes required to synthesize six amino acids—glutamate, glutamine, alanine, aspartate, cysteine and glycine. In addition, the genome encodes a partial set of enzymes—such as phosphoglycerate dehydrogenase, phosphoserine phosphatase, and acetolactate synthase—for producing another five amino acids (Table 1). Among the complete pathways, we identified nine specific enzyme-coding genes, including those for glutamine synthetase, alanine transaminase, and cystathionine-β-synthase, all of which were highly expressed in all instars, suggesting that *E. pela* is capable of self-generating these amino acids. Of note, glutamic acid and aspartate are important precursors for the synthesis of other amino acids, even though *E. pela* cannot independently generate those amino acids.

In contrast to aphids and other insects [5,31], we did not find *Buchnera* and other common symbionts that provide amino acids in *E. pela*. However, we did find two fungi in the genus *Ophiocordyceps*, *O. sinensis* and *O. incosistent* (Figure 5A). Both of these fungal species were highly abundant in the nymphs, pupae, and female adults (Figure 5B). To further localize the *Ophiocordyceps* fungus in the insect body, we synthesized a specific probe labeled with red fluorescent protein according to the characteristic ITS sequence of *O. sinensis*, and performed immunofluorescence hybridization with ultra-thin paraffin sections of *E. pela* tissues. This assay clearly revealed the location of *O. sinensis* in the fat bodies of *E. pela* (Figure 5C), which aligns with the important role of the fat body as a nutrition-producing organ.

To determine if *O. sinensis* is capable of producing those essential amino acids that *E. pela* cannot self-produce and which must be acquired from outside sources, we conducted the metagenomic analysis of *O. sinensis*. We found that 62 enzymes of *O.sinensis* responsible for synthesizing all twenty amino acids were actively expressed in both nymphs and adults of *E. pela* (Figure 5D), indicating *O. sinensis* can synthesize the complete panel of amino acids as required for protein generation. Taken together, our data are consistent with a model in which *E. pela* harvest essential amino acids from symbiotic *Ophiocordyceps* fungi.

### 3.5. Model of Nitrogen Cycle in E. pela

Based on the above evidence, we present a nitrogen cycling model for *E. pela* (Figure 6) comprising the following sequential processes: (1) atmospheric nitrogen (N_2_) enters the insect’s respiratory system through spiracles; (2) N_2_ diffuses into the insect’s tracheal system and is subsequently transported via tracheal capillaries to fat body cells; (3) within these fat body cells, symbiotic *Rhizobiales* and *Methylobacterium* strains mediate enzymatic nitrogen fixation, converting N_2_ to ammonia (NH_3_); (4) the resulting NH_3_ undergoes hydration to form ammonium ions (NH_4_^+^), serving as a critical precursor for amino acid biosynthesis.

Our model delineates two distinct amino acid production pathways in *E. pela*.

Primary pathway: NH_4_^+^ is actively translocated via ammonia-specific transporters to endosymbiotic *O. sinensis* residing in the fat body. The fungal partner synthesizes amino acids through its biosynthetic machinery, subsequently exporting these nitrogenous compounds to fulfill the host’s nutritional requirements.

Ancillary pathway: A subpopulation of NH_4_^+^ is directly assimilated by *E. pela*’s endogenous metabolic pathways to synthesize six essential amino acids.

These complementary nitrogen assimilation mechanisms demonstrate functional redundancy, ensuring metabolic flexibility and sustained nitrogen provision under varying nutritional conditions. The coordinated interplay between microbe and host metabolic capacity establishes an evolutionarily optimized strategy for nitrogen resource utilization in this specialized insect–fungus symbiotic system.

## 4. Discussion

Each insect species harbors a unique set of microbial species [32,33]. Symbiotic microorganisms in phloem-sap-feeding insects (PSFIs) have formed a closed relationship over evolutionary time. Most of the PSFIs, like aphids, can feed on different parts of plants or between different plants to obtain nutrients. Scale insects are overly dependent on the host plant, and for most of their lifetime they are attached to the host and unable to move for feeding. The sessile lifestyle of *E. pela* resulted from a unique relationship with symbiotic microorganisms, which allows it to fix nitrogen from the air and synthesize essential amino acids by the fungus *O. sinensis*. This unique nitrogen-recruiting system represents a remarkable evolutionary adaptation that effectively compensates for the inherent nitrogen deficiency in phloem sap, ensuring the insect’s nutritional requirements are met despite its immobile lifestyle.

In addition, we find that *E. pela* exhibits stage-specific variations in nitrogen fixation and amino acid synthesis, reflecting its distinct biological characteristics at different developmental stages. Notably, the abundance of two dominant nitrogen-fixing bacteria, *Rhizobiales* and *Methylobacterium*, peaks in secondary instar nymphs. However, the highest expression levels of nitrogen fixation activity and amino acid synthases are observed in adult stages, which represents the longest developmental phase in the *E. pela* life cycle, extending from autumn to early summer of the following year. This period coincides with the reduced photosynthetic activity in the host plant [34] (Appendix A). The enhanced nitrogen fixation and amino acid synthesis activities of symbiotic microorganisms in adult *E. pela* potentially function as a strategy to counter nitrogen deficiencies in the host plant sap, which ensures adequate nitrogen acquisition during periods of limited host plant nutritional availability.

The capacity for nitrogen fixation in insects has been well-documented in termites and beetles [13,14,15,16,17]. In these systems, microbial populations must be re-established each generation through dietary acquisition [33,34], resulting in bacterial communities that are primarily restricted to the intestinal tract. Consequently, these associations represent facultative symbioses [17,18] that are inherently unstable and significantly influenced by dietary composition [33,34]. In contrast to these transient associations, *E. pela* maintains an obligate symbiotic relationship with vertically transmitted microorganisms. This evolutionary adaptation likely reflects the sessile nature of *E. pela*, which limits opportunities for horizontal microbial acquisition through environmental exposure. Atmospheric nitrogen fixation provides a reliable and consistent nutritional strategy for *E. pela*, but necessitates the vertical transmission of symbionts to ensure adequate microbial colonization in successive generations. This transmission mechanism represents an evolutionary adaptation to the host’s immobile lifestyle, ensuring consistent symbiont acquisition despite limited environmental contact.

Notably, via high-throughput sequencing analysis, bacteria of the family Rhizobiaceae have been detected not only in *E. pela*, but also across 19 species of Coccidae, including *Eucalymnatus tessellatus* (tessellated cycad scale), *Dicyphococcus ficicola* (banyan double-horned wax scale), and species of the genus *Saissetia*. Among these, *Rhizobiales* bacteria emerge as the most dominant microbial taxa, with their relative abundance consistently exceeding 40% in *Eucalymnatus tessellatus* and *Saissetia* species [35]. These findings suggest that in Coccidae insects—characterized by reduced locomotive capacity and profound reliance on host-derived nutrients—compensating for their nitrogen-deficient diet through nitrogen fixation by symbiotic bacteria may represent a widespread adaptive strategy.

Fossil evidence indicates that Archaeococcoidea inhabited terrestrial environments, subsisting on fallen leaves, low-lying vegetation, fungi, and other organic substrates [36] During the Cretaceous period, the proliferation of gymnosperms and angiosperms expanded food resource availability, facilitating the transition of scale insects from soil-dwelling to above-ground plant habitats [37]. Concurrently, insect lineages diversified their mouthpart structures to exploit a broader range of food resources [38], with scale insects evolving specialized mouthparts adapted for phloem sap extraction. Phloem sap is characterized by high concentrations of energy-rich sugars, but is nitrogen-poor. *Rhizobiales*, which exhibit a strong affinity for sugary substrates [39], rely on phloem sap as a stable source of sugars and water. Within eukaryotic cells, the plasma membrane creates an anaerobic microenvironment critical for nitrogen fixation, shielding nitrogenase enzymes from oxidative damage. Ancestral scale insects likely acquired nitrogen-fixing bacteria and *Ophiocordyceps* from the soil, establishing an obligate symbiotic association. This mutualism compensated for the nitrogen deficit in phloem sap through the partners’ nitrogen fixation and amino acid synthesis capabilities, enabling scale insects to progressively occupy a nutritional niche where phloem sap serves as the primary resource. How exactly this remarkable symbiotic relationship initially arose remains to be further studied.

## 5. Conclusions

This study revealed *E. pela* obligate symbiosis with nitrogen-fixing bacterial groups *Rhizobiales* and *Methylobacterium* and amino-acid-synthesizing *Ophiocordyceps*, compensating for the low-nitrogen phloem sap. Stage-specific analysis showed heightened nitrogen fixation in adults, matching host nutritional limitation. Cross-taxa comparisons identified that *Rhizobiales* have been detected in at least 19 other scale insect species, indicating this symbiotic strategy is widespread in Coccidae. Unlike facultative symbioses (e.g., termites, beetles) reliant on horizontal microbial acquisition, scale insects’ vertical transmission ensures stable symbiont colonization, thus adapting to their sessile lifestyle. This highlights symbiosis as critical for overcoming dietary nitrogen limitation across scale insects.

## Figures and Tables

**Figure 1 insects-16-00836-f001:**
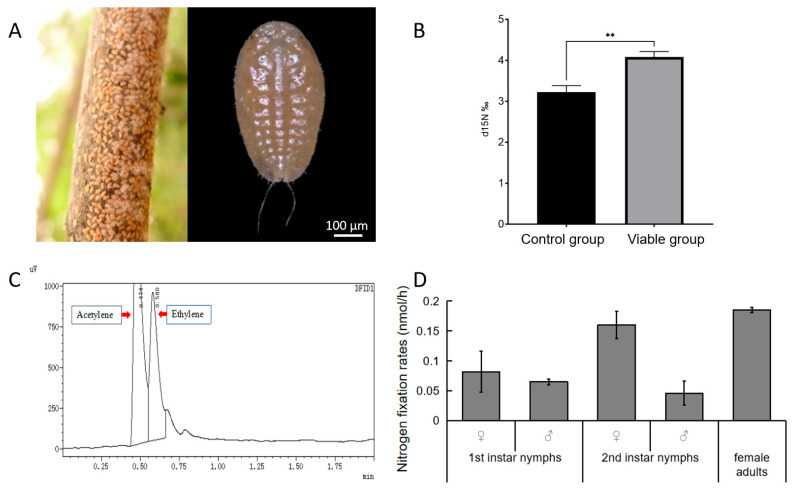
Nitrogen fixation ability of *E. pela*. (**A**) Nymphs of *E. pela* aggregating on branches. (**B**) Quantification of ^15^N content reveals that viable controls (left) exhibit a significantly higher ^15^N content than ethanol-treated controls (right) following exposure to isotope-labeled ^15^N nitrogen gas (*p* < 0.01 (**)). (**C**) Gas chromatography (GC) traces showing acetylene and ethylene peaks during the acetylene reduction assay. (**D**) Nitrogen fixation rates of *E. pela* across different life stages.

**Figure 2 insects-16-00836-f002:**
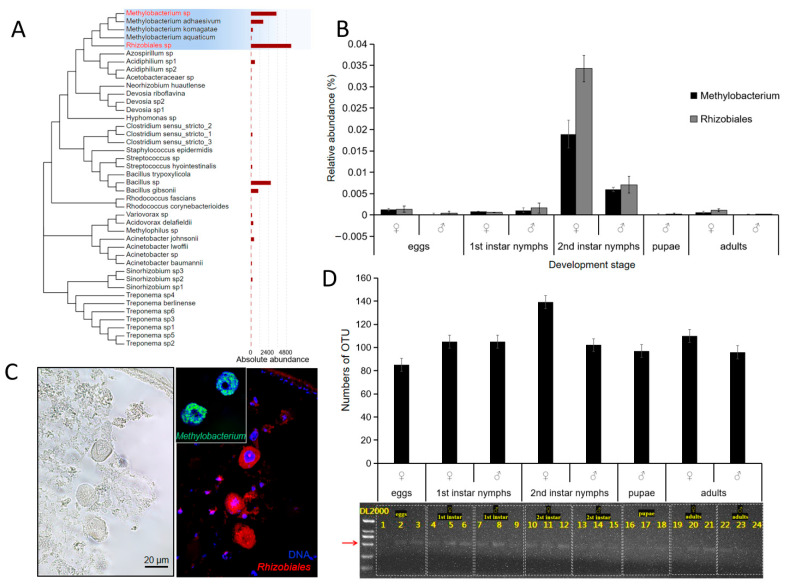
Nitrogen-fixing bacteria in *E. pela*. (**A**) Forty-two nitrogen-fixing bacterial taxa in *E. pela*. (**B**) The relative abundances of nitrogen-fixing bacteria in different development stages. (**C**) Immunofluorescence localization demonstrating the presence of *Rhizobiales* and *Methylbacterium* within fat body cells. (**D**) The *nifH* gene was detected in all developmental stages of *E. pela*. The red arrow points to the amplified *nifH* band (723 bp).

**Figure 3 insects-16-00836-f003:**
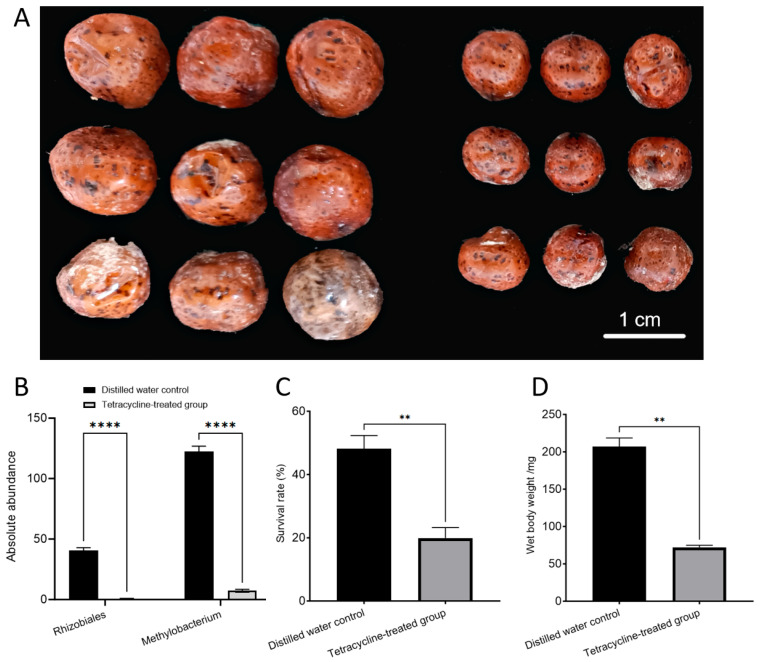
Effects of removing nitrogen-fixing bacteria in female *E. pela*. (**A**) Size difference between female adult treated with distilled water (left) and tetracycline (right). (**B**) The absolute abundance of nitrogen-fixing bacteria in female adult *E. pela* treated with distilled water was significantly higher than that in those treated with tetracycline (*p* < 0.0001 (****)). (**C**) The survival rate of females without removal of symbiotic bacteria was significantly higher than that in females with removal of symbiotic bacteria (*p* < 0.01 (**)). (**D**) The individual weights of females without removal of symbiotic bacteria were significantly higher than that in females with removal of symbiotic bacteria (*p* < 0.01 (**)).

**Figure 4 insects-16-00836-f004:**
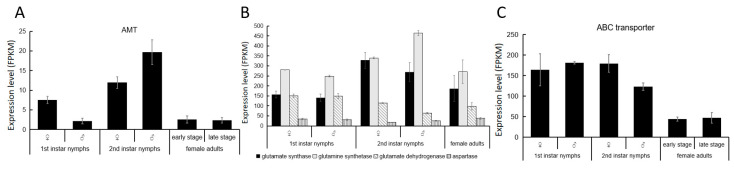
The amino acid biosynthesis pathway in *E. pela*. (**A**) Expression level of AMT in different developmental stages. (**B**) Key enzymes in the post-transamination amino acid biosynthesis pathway. (**C**) Expression levels of ABC transporter gene in different developmental stages. Data are presented as the mean ± standard deviation (SD, *n* = 3).

**Figure 5 insects-16-00836-f005:**
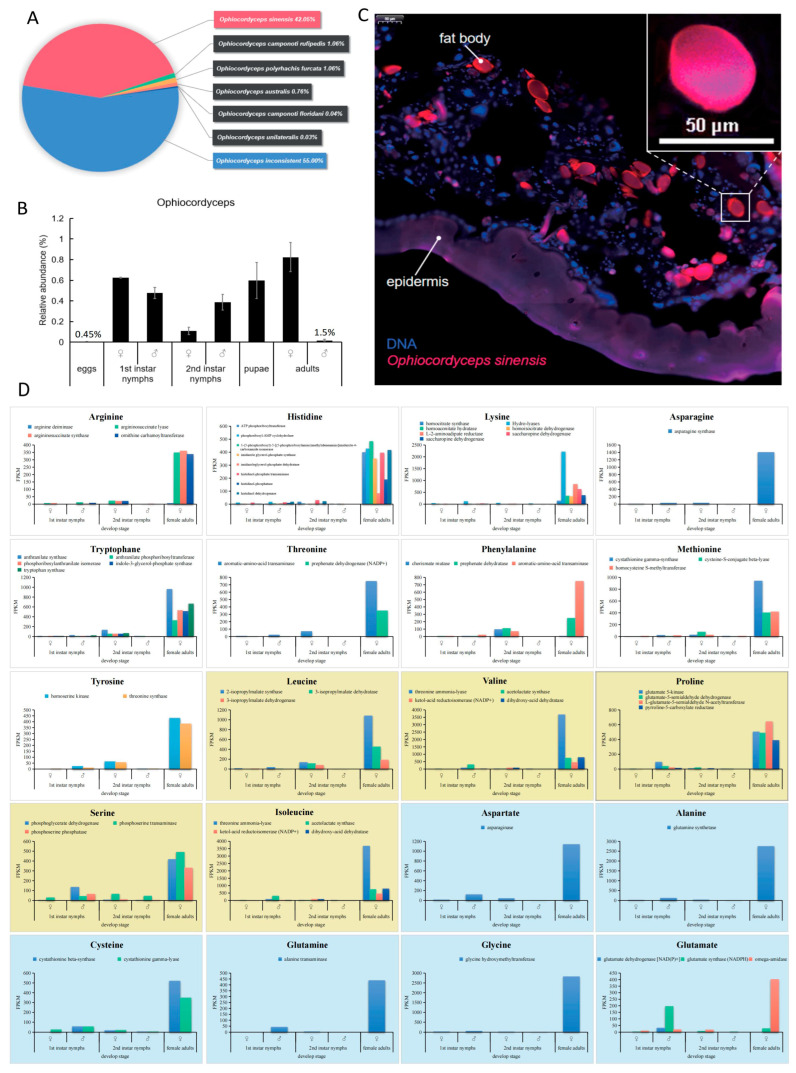
The abundance and distribution of *Ophiocordyceps* in *E. pela*. (**A**) Taxonomic composition of seven *Ophiocordyceps* species detected in *E. pela*. (**B**) Dynamic changes in *Ophiocordyceps* relative abundances across larval–pupal–adult developmental stages. (**C**) Localization of *Ophiocordyceps sinensis* in host fat body tissues. (**D**) Expression levels of enzyme genes involved in the amino acid synthesis of *O.sinensis* and *E. pela*. The expression patterns of 62 enzyme-coding genes involved in 20 essential amino acid synthesis pathways in different instars of *O. sinensis*. White boxes: Amino acids not synthesized by *E. pela* (complete metabolic dependence). Yellow boxes: Partial enzymatic capacity in *E. pela* (symbiosis-dependent biosynthesis). Blue boxes: Amino acids fully synthesized by *E. pela* (host-autotrophic production). Data are presented as the mean ± standard deviation (SD, *n* = 3).

**Figure 6 insects-16-00836-f006:**
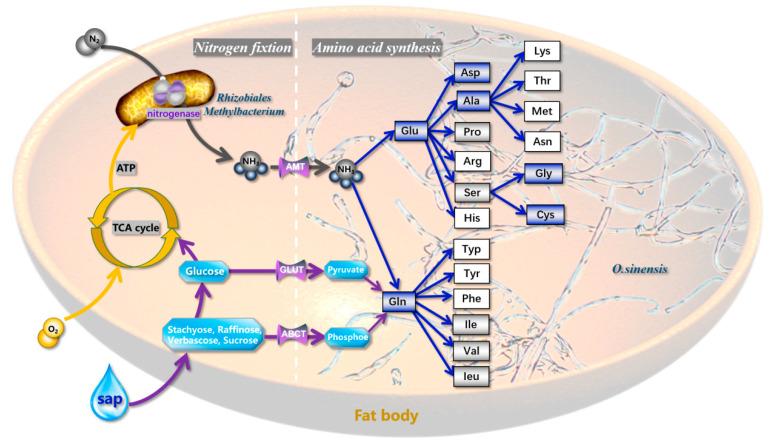
Nitrogen cycling and amino acid biosynthesis in *Ophiocordyceps*–microbe system. Symbiotic nitrogen fixation: *Rhizobiales* and *Methylobacterium* biologically fix atmospheric nitrogen in host fat bodies. Ammonium assimilation: NH_3_ hydrolysis to NH_4_^+^ (water phase) catalyzes amino acid biosynthesis by *O. sinensis*. Host contribution: The catabolism of phloem sap sugars generates metabolic energy to support nitrogen fixation and amino acid biosynthesis. Blue frames: Amino acids fully synthesized by *E. pela*. Gray frames: Partial enzymatic pathways retained in *E. pela*. White frames: No functional enzymes detected for these amino acids (The blue boxes in the figure represent amino acids that E. pela can synthesize independently; the gray boxes denote amino acids for which the insects can only encode a subset of the enzymes in their synthetic pathways; and the white boxes indicate amino acids that the insects are completely unable to synthesize).

**Table 1 insects-16-00836-t001:** Eight enzymes of six amino acids’ synthesis and five amino acids’ part synthesis in *E. pela*.

Category	Amino Acid	Enzymes	EC NO.	Chromosome	Strand	Start	End	Length
WSI synthesized amino acids	Glutamate	glutamate dehydrogenase glutamate synthase omega-amidase	1.4.1.3 1.4.1.13 3.5.1.3	fragScaff_scaffold_49_pilon fragScaff_scaffold_128_pilon original_scaffold_845_pilon	- + +	35,092 213,176 267,067	67,466 266,180 271,664	1650 6081 2042
	Glutamine	glutamine synthetase	6.3.1.2	original_scaffold_1706_pilon	+	10,643	67,949	2654
	Aspartate	L-asparaginase	3.5.1.1	fragScaff_scaffold_86_pilon	+	662,258	716,055	3847
	Alanine	alanine transaminase	2.6.1.2	fragScaff_scaffold_59_pilon	+	701,946	712,045	3426
	Cysteine	cystathionine-β-synthase cystathionine-γ-lyase	4.2.1.22 4.4.1.1	fragScaff_scaffold_162_pilon fragScaff_scaffold_432_pilon	+ +	268,305 48,151	296,812 72,673	1644 1293
	Glycine	serine hydroxymethyl transferase	2.1.2.1	fragScaff_scaffold_4_pilon	+	3,494,937	3,500,594	1419
WSI partially synthesized amino acids	Serine	phosphoglycerate dehydrogenase phosphoserine aminotransferase	1.1.1.95 2.6.1.52	fragScaff_scaffold_536_pilon -	+ -	488 -	10,160 -	1344 -
		phosphoserine phosphatase	3.1.3.3	fragScaff_scaffold_40_pilon	+	794,412	797,910	666
	Proline	glutamate 5 kinase L-glutamate-5-semialdehyde N-acetyltransferase	2.7.2.11 2.3.1.271	fragScaff_scaffold_263_pilon -	- -	1,276,602 -	1,332,893 -	1726 -
		pyrroline-5-carboxylate reductase	1.5.1.2	fragScaff_scaffold_32_pilon	+	1,175,590	1,182,674	1536
	Valine/ Isoleucine	threonine dehydratase ketol-acid reductoisomerase	4.3.1.19 1.1.1.86	fragScaff_scaffold_277_pilon -	- -	2,902,504 -	2,933,483 -	1296 -
		dihydroxy-acid dehydratase	4.2.1.9	-	-	-	-	-
		acetolactate synthase	2.2.1.6	fragScaff_scaffold_78_pilon	-	104,891	111,100	3064
	Leucine	amino acid aminotransferase	2.6.1.42	fragScaff_scaffold_21_pilon	+	18, 138	43,772	2512

## Data Availability

The original contributions presented in this study are included in the article. Further inquiries can be directed to the corresponding authors.

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
