# Peer review of "Ecological Significance of a Novel Nitrogen Fixation Mechanism in the Wax Scale Insect Ericerus pela"

_insects, 2025, doi:10.3390/insects16080836_

Round 1
Reviewer 1 Report
Comments and Suggestions for Authors
1. The manuscript under review is extremely relevant, devoted to an urgent task - the search for mechanisms of nitrogen fixation in insects feeding on low-nitrogen food. This group includes a large number of phytophagous and xylophagous insects, which are important for agriculture and forestry. Therefore, knowledge of the mechanisms of their nutrition and reproduction can be important for the development of methods for population control.
2. The introduction fully reflects the essence and relevance of the work, however, it is necessary to clearly formulate the purpose of the study, since lines 97-108 provide the planned result and the scheme of experiments. It is necessary to clearly rephrase.
3. Materials and methods are well described and reflect all experiments and their processing.
Note. It is necessary to indicate how many copies of E. Pela were used for DNA extraction, did the insects go through starvation to remove excrement from the intestines? How did the differentiation of ectosymbionts from endosymbionts that participate in digestion occur?
4. The results of the work are well described and statistically processed. However, the captions and legends to the figures need to be made larger to facilitate the perception of the readers.
5. The "Discussion" section is written in a very abbreviated form. The authors have identified important physiological moments for phytophagous insects, however, this section lacks data and their discussion on other groups of insects. At the same time, the "conclusion" provides ambitious assumptions about the adaptive evolution of scale insects. What data are these conclusions based on?
6. The "Conclusion" section needs to be rewritten from the particular to the general, based on the results of the studies obtained. Evolutionary questions remain unproven.
Author Response
Thank you very much for taking the time to review this manuscript. Please find the detailed responses below and the corresponding revisions/corrections in track changes in the re-submitted files.
Point-by-point response to Comments and Suggestions for Authors
Comments 1: The manuscript under review is extremely relevant, devoted to an urgent task - the search for mechanisms of nitrogen fixation in insects feeding on low-nitrogen food. This group includes a large number of phytophagous and xylophagous insects, which are important for agriculture and forestry. Therefore, knowledge of the mechanisms of their nutrition and reproduction can be important for the development of methods for population control.
Response 1: We sincerely appreciate the Reviewer for recognizing the relevance and urgency of our work, and fully agree that elucidating nitrogen fixation mechanisms in these ecologically and agriculturally critical insects holds significant promise for advancing both fundamental science and pest management strategies.
Comments 2: The introduction fully reflects the essence and relevance of the work, however, it is necessary to clearly formulate the purpose of the study, since lines 97-108 provide the planned result and the scheme of experiments. It is necessary to clearly rephrase.
Response 2: Thank you for pointing this out. We agree with this comment. Therefore, I have rewrote this paragraph of the introduction, this change can be found page 3, paragraph 1, and line 101-115.
Comments 3: Materials and methods are well described and reflect all experiments and their processing. Note. It is necessary to indicate how many copies of E. Pela were used for DNA extraction, did the insects go through starvation to remove excrement from the intestines? How did the differentiation of ectosymbionts from endosymbionts that participate in digestion occur?
Response 3: Thank you for your questions.
â‘ For DNA and RNA extraction, the following quantities of E. pela were used: 100 1st-instar male and female nymphs; 30 2nd-instar male and female nymphs, 30 male pupae, and 30 male adults; 10 female adults. Each group included 3 biological replicates to ensure statistical robustness. And I have added these details in the materials and methods section of the manuscript ( page 3, line 131-134).
â‘¡ Starvation Treatment for Gut Content Removal: 1st-instar nymphs were collected immediately after hatching from eggs, with no prior contact with host plants or feeding, ensuring minimal gut content contamination. Pupae and male adults were collected by peeling off their wax shells and similarly subjected to no feeding prior to sampling, maintaining gut sterility. 2nd-instar nymphs and female adults : Due to their strong attachment to host plants, samples were quickly desiccated and died upon detachment. Thus, they were used directly without starvation, as prolonged handling would compromise sample integrity. I have added these details in the materials and methods section of the manuscript ( page 5, line 219-228).
â‘¢ Ectosymbionts and endosymbionts were not initially distinguished in the sequencing experiments. Instead, genomic DNA was extracted directly from whole insects following preliminary surface cleaning (each sample was washed 3 times with 75 % alcohol and undergo ultrasonic cleaning 3 times with distilled water). Further differentiation between ectosymbionts and endosymbionts was achieved via FISH assays. Fluorescent probes targeting conserved 16S rRNA sequences of Ophiocordyceps, Rhizobiales, and Methylobacterium were used, revealing that they were localized exclusively within the insect fat body. This spatial localization confirmed their status as endosymbionts.
Comments 4: The results of the work are well described and statistically processed. However, the captions and legends to the figures need to be made larger to facilitate the perception of the readers.
Response 4: Agree. I have increased the font size of the captions and legends for all figures in the text and added more detailed descriptions to the figures to facilitate reader comprehension.
Comments 5: The "Discussion" section is written in a very abbreviated form. The authors have identified important physiological moments for phytophagous insects, however, this section lacks data and their discussion on other groups of insects. At the same time, the "conclusion" provides ambitious assumptions about the adaptive evolution of scale insects. What data are these conclusions based on?
Response 5: Thank you for pointing this out. We agree with your comment, so we have revised the Discussion section. Specifically: We have added discussions on other insect groups, including comparisons between E. pela and other piercing-sucking mouthpart insects (e.g., aphids) (page 15, line 502-509), comparisons of nitrogen fixation mechanisms between E. pela and other nitrogen-fixing insects (page 15, line 524-538), and evidence for the detection of nitrogen-fixing bacteria in various scale insects (page 15, line 539-548). We have added explanation for the hypothesis of adaptive evolution of scale insects (page 16, line 549-566).
Comments 6: The "Conclusion" section needs to be rewritten from the particular to the general, based on the results of the studies obtained. Evolutionary questions remain unproven.
Response 6: Yes, we agree with your point, so we have rewritten the conclusion section to avoid over-inference. This change can be found page 16, paragraph 2, and line 568-578.
Reviewer 2 Report
Comments and Suggestions for Authors
Authors present an interesting novel way of nitrogen fixation and amino acids syntheses in a phloem-sap feeding scale insect. This is highly innovative and the results are convincing, but the manuscript needs revision.
Majors:
- Several methods used in the study are not described in the Material and Methods part (qRT-PCR, survival rates, amount of antibiotics used and others)
- Fig. 1A is not mentioned in the text
- Figure legend do no say, what bars in the figures show (SD, SEM?, means of how many values?)
- Legend to Fig. 1B: left and right are changed. Only those significance values shown in the figure should be explained
- Legend to Fig. 5: there are no yellow boxes
Minors:
- the manuscript needs very careful check for spelling/typing errors
- insert a space between numbers in units
- check use of lowercase and uppercase letters in headings/subheadings
- write chemicals used in lowercase letters
- pH, 16S etc. is correct
- use either l or L for liter, but not mixed
- use full sentences in Materials and Methods and do not begin a sentence with a number
- give species names in the References in italics
Author Response
Thank you very much for taking the time to review this manuscript. Please find the detailed responses below and the corresponding revisions/corrections in track changes in the re-submitted files.
Point-by-point response to Comments and Suggestions for Authors
Comments 1: Several methods used in the study are not described in the Material and Methods part (qRT-PCR, survival rates, amount of antibiotics used and others).
Response 1: Thank you for raising this point. To address the concern, we have now added detailed descriptions of the experimental methods mentioned (qRT-PCR, survival rates, antibiotic usage, and related procedures) to the Materials and Methods section. Specific revisions are highlighted below for easy reference:
â‘ qRT-PCR: Detailed protocols are now included in Section 2.9 (page 6, paragraph 2, lines 267-282).
â‘¡Survival rates and antibiotic dosage: Methodological details have been added to Section 2 .10 (page 7, paragraph 2, lines 289-301).
â‘¢Additionally, we have further elaborated on other relevant technical details of the experimental methods to ensure comprehensiveness.
Comments 2: Fig. 1A is not mentioned in the text
Response 2: Thank you for pointing this out. I inserted the description of Fig. 1A. This change can be found page 7, paragraph 3, and line 303-306.
Comments 3: Figure legend do no say, what bars in the figures show (SD, SEM?, means of how many values?)
Response 3: Thank you for pointing out the issue. The bars in the figure denote the standard deviation (SD), with 3-5 biological replicates per group. To enhance clarity, I have incorporated a more detailed figure legend into the manuscript, these revisions are located at: pages 8 (lines 331-333), page 9 (line 371), page 11 (lines 392-394), page 12 (line 438-439), and page 17 (lines 470-471).
Comments 4: Legend to Fig. 1B: left and right are changed. Only those significance values shown in the figure should be explained.
Response 4: Thanks for pointing it out, We have checked and updated Fig. 1B. This change can be found page 8 (line 325).
Comments 5: Legend to Fig. 5: there are no yellow boxes
Response 5: Thank you for pointing this out. We have revised the Fig. 5, with the color in the figure corrected to yellow. This change can be found page 13 (line 460).
Comments 6: the manuscript needs very careful check for spelling/typing errors
insert a space between numbers in units
check use of lowercase and uppercase letters in headings/subheadings
write chemicals used in lowercase letters
pH, 16S etc. is correct
use either l or L for liter, but not mixed
use full sentences in Materials and Methods and do not begin a sentence with a number
give species names in the References in italics
Response 6: Thank you for pointing out these issues. We agree that attention to detail is critical, and we have now conducted a full review of the entire manuscript to correct these problems and ensure they do not recur.
Round 2
Reviewer 1 Report
Comments and Suggestions for Authors
All the comments and suggestions were satisfactorily addressed and the manuscript can be accepted for publication.